# An Analysis of Health Perceptions and Performance in Elementary Students in Korea during the ongoing COVID-19 Pandemic

**DOI:** 10.3390/healthcare11010083

**Published:** 2022-12-27

**Authors:** Yongsuk Seo, Chul-min Kim, Min-jun Kim, Eui-jae Lee, Hyun-su Youn

**Affiliations:** 1Sports AIX Graduate Program, Pohang University of Science and Technology, Pohang 37673, Republic of Korea; 2Department of Physical Education, College of Education, Korea University, Seoul 02841, Republic of Korea; 3Department of Physical Education, Shin Han University, Uijeongbu 11644, Republic of Korea; 4Department of Physical Education, Graduate School of Education, Sogang University, Seoul 04107, Republic of Korea; 5Department of Physical Education, College of Education, WonKwang University, Iksan-si 54538, Republic of Korea

**Keywords:** elementary student, health perception, importance-performance analysis (IPA), ongoing COVID-19 pandemic

## Abstract

During the coronavirus disease 2019 (COVID-19) pandemic, social distancing guidelines changed lifestyles, including increased sedentary time, physical inactivity, and disrupted sleep patterns among children. The purpose of the present study is to analyze the health awareness (mental health, disease, physical activity, sleep, eating habit, and hygiene health management) of elementary school students during the COVID-19 pandemic, and use the importance-performance analysis (IPA) technique to identify gender differences in health perceptions. We collected data on 1006 students, which was analyzed using frequency analysis, reliability testing, independent sample t-tests, and importance-performance analysis (IPA). A median importance value of 0.163 and a median performance value of 4.048 were selected as cross points to distribute the IPA matrix into four quadrants. The highest performance was given for wearing a mask and sanitary practice; the IPA matrix indicated that the sense of belonging, happiness, trust, and movement activity were located in quadrant I. Children’s regular physical activity and level of physical activity were low, especially that of girls. Children’s sleep management was poor. Their physical activity and sleep-related factors must be improved under the facilitation of the national government, public education institutions, and families.

## 1. Introduction

Coronavirus disease 2019 (COVID-19), caused by the severe acute respiratory syndrome coronavirus 2 (SARS-CoV-2), is easily transmitted. Consequently, the World Health Organization (WHO) declared the novel coronavirus (COVID-19) outbreak a global pandemic on 11 March 2020 [1]. As of 21 June 2021, more than 170 million confirmed cases and 3.8 million deaths have been reported to the WHO Dashboard globally [2]. The global spread of COVID-19 has led to social distancing, lockdowns, and the use of face masks. Despite social distancing guidelines, the COVID-19 virus has mutated, and multiple vaccines against SARS-CoV-2 have been developed or are being developed worldwide over the past year [2].

During the COVID-19 pandemic, many countries implemented school closure as part of social distancing to control transmission of infection. The social distancing guidelines have changed lifestyles, increased sedentary time and physical inactivity, and disrupted sleep patterns [1]. These issues are pertinent to both adults and children; they negatively impact child and adolescent physical and psychological health, and cause depression, anxiety, social isolation, maladjustment, and stress [3,4]. Interestingly, the COVID-19 outbreak has brought global attention to the importance of health [5]. However, statistically significant differences have been observed between the perceived importance and performance of mental health, disease, physical activity, sleep, and diet management during the COVID-19 pandemic [6]. In particular, among Korean middle-school students, the level of physical activity had the lowest importance and performance.

Interestingly, the COVID-19 outbreak brought attention to the importance and value of health across the globe [5]. However, a previous study indicated that there were statistically significant differences between the perceived importance and performance of mental health, disease, physical activity, sleep, and diet during the COVID-19 pandemic [6]. In particular, the level of physical activity had the lowest importance and performance among Korean middle school students.

Although several studies have determined the health awareness of adolescents during COVID-19 using IPA, few studies have examined the gender difference between importance and performance in pre-adolescents (aged 10–13 years) in Korea.

Elementary school is the starting point of education in understanding the importance of cultivating a healthy body and mind and developing practical skills. It is a crucial time to identify problems highlighted by the COVID-19 pandemic and activate health activities. To this end, education that can cultivate life skills through the practice of health activities in school and daily life is necessary. Therefore, this study aims to help implement elementary physical education that can prevent students’ health risk behaviors to enable the achievement of individual goals and ensure their contribution to the nation as healthy democratic citizens.

On 2 May 2022, the Korean government announced that the mandate to wear masks outdoors will be adjusted and it will not be required for gatherings of 50 or fewer people. With these guidelines, school sports club activities and after-school activities were reactivated.

Therefore, the study determines the difference between the importance and performance of health perceptions in pre-adolescents (aged 10–13 years). It specifically explores the current state of children’s health since mandatory outdoor mask use regulations have been lifted. The specific aims were to (1) analyze the difference between the importance and performance of health perception, in terms of mental health, disease, physical activity, sleep, dietary habits, and hygiene management, in pre-adolescents. (2) Examine gender differences in perceived importance and performance of health. The data could be valuable for future planning and implementation of public and private educational institutions in health education in the post-COVID-19 era.

## 2. Material and Methods

### 2.1. Participants

The present study started in September 2022. In October 2022, students from the third to sixth grades (aged 10–13 years) of an elementary school in Korea were selected as research participants using convenience sampling to understand health awareness and performance. Online (Google Form) and offline surveys were conducted through teachers at Korean elementary schools. The study was conducted after obtaining approval from the Institutional Review Board of Wonkwang University (WKIRB-202209-SB-081), and the participants of this study voluntarily signed the written informed consent form.

### 2.2. Instruments

This study employed convenience sampling, a non-probability sampling method. We utilized a nominal scale that was deemed appropriate, and the modified Ware’s health perception scale [7], which has been verified by previous studies [8,9,10]. The latter has been modified for the current study. The questionnaire was designed to assess six specific categories of health perception: mental health, disease, physical activity, sleep, dietary habits, and hygiene management. Participants were provided with 40 questions and instructed to choose a number between 1 and 5, with 1 being (strongly disagree) and 5 being (strongly agree). These questions were associated with the six categories of health perception, and each category was calculated independently.

This study utilized an important performance analysis (IPA). IPA has been used in various fields because it is a simple and useful technique for schematizing survey results and establishing efficient strategies [11,12]. The IPA technique evaluates the importance and performance and displays the results on the *X*-(importance) and *Y*-(performance) axes. The results are presented in the quadrant of the IPA matrix [13].

The perceived importance and performance of each variable were evaluated with a modified IPA because it is effective for integrated partial correlation analysis and natural logarithmic transformation which measures the importance of attributes [14,15]. The average values of relative importance and performance in the grids of the *X*- and *Y*-axes are expressed as grids.

### 2.3. Reliability of Instruments

Varimax, an orthogonal rotation method, was used to verify the construct validity of the survey tool used in this study, and Cronbach’s α was used to verify its reliability. The factor-loading values for each factor was 0.5 or more, and the total explanatory power was 67.756%, and six factors were extracted. In addition, Cronbach’s α was at least 0.720 and at most 0.771, and the standard value was 0.6 or higher, showing internal consistency [16]. There are details in Table 1.

### 2.4. Procedure and Statistical Analysis

Using a statistical software package (SPSS v. 18.0), a frequency analysis was conducted to determine the demographic characteristics of the participants (sex and age). Exploratory factor and reliability analysis Cronbach’s α coefficient were used to verify the validity and reliability of the survey tool. Furthermore, an independent sample *t*-test was performed to determine the difference between the sexes. A revised IPA method was used to verify the importance and performance of each factor in all children.

## 3. Results

### 3.1. Participants’ Demographic Character

To understand the health awareness and performance of Korean teenagers (aged 10–13 years), we obtained data from 1006 participants (481 boys and 525 girls) through online and offline surveys via elementary school teachers. Table 2 shows the demographic characteristics of the participants.

### 3.2. Analysis of the Importance and Performance of Each Factor

Table 3 shows the results of IPA, indicating that the average importance was 0.163, and the average performance was 4.048.

In the performance rating scale, “wearing a mask” and “practicing hygiene” were rated 4.5 or higher, followed by “a sense of belonging”, “trust”, “prescription management”, “regular eating activity”, “movement activity”, “practicing distance”, and “disease prevention” with a rate of 4.0 or higher. Additionally, 3.5 or higher was the rate for adequate quantity of food, sports activities, sleep environments, meal habits, sleep hygiene, and regular physical activity. “Regular sleep” had the lowest performance rate.

### 3.3. Analysis of the Importance and Performance Matrix

A median importance value of 0.163 and a median performance value of 4.048 were selected as cross points to distribute the IPA matrix into four quadrants (Table 4); (Figure 1).

In quadrant I (“Keep up the good work”), indicating high importance and high performance, a sense of belonging, happiness, trust, and physical movement were placed. In quadrant II (“concentrate here”), indicating high importance and low performance, adequate quantity of food, dietary habits, sports activity, and regular physical activity were placed. In quadrant III, (“low priority”), indicating low importance and low performance, preventive care, sleep environment, sleep hygiene, and regular sleep were located. In quadrant IV (“possible overkill”), indicating low importance and high performance, social distancing, prescription control, and disease prevention were placed.

### 3.4. Difference in Performance between Boys and Girls

Table 5 shows a comparison of the performance of boys and girls. An independent sample *t*-test revealed that physical activity was significantly higher in boys (4.028 ± 0.826) than in girls (3.744 ± 0.901) (*t* = 5.221, *p* ≤ 0.001).

### 3.5. Analysis of the Importance and Performance Matrix between Boys and Girls

For boys, a median importance value of 0.160 and a median performance value of 4.072 were classified as the cross points for the IPA matrix analysis. For girls, the median importance value of 0.170 and median performance value of 4.026 were used as the cross points (Table 6 and Table 7) (Figure 2).

In quadrant I, happiness, trust, sense of belonging, physical movement, and sports activity were placed for boys, while happiness and social distancing were placed for girls. In quadrant II, regular physical activity was observed in boys, while regular physical activity, movement activity, sports activity, dietary habits, and adequate meals were observed in girls. In quadrant III, for boys, sleep environment, sleep hygiene, regular sleep, disease prevention, preventive care, adequate diet, and dietary habits were located, whereas sleep environment, sleep hygiene, and regular sleep were located for girls. In quadrant IV, wearing a mask, hygiene practice, prescription control, social distancing, and regular meals were placed for boys, while wearing a mask, sanitary practice, prescription control, disease control, sense of belonging, and trust were located for girls. The IPA matrix indicated that happiness in quadrant I, regular physical activity in quadrant II, sleep environment, sleep hygiene, regular sleep in quadrant III, and wearing a mask, sanitary practice, and prescription care in quadrant IV were located for both boys and girls.

## 4. Discussion

This study analyzed the importance and performance of health perceptions during the ongoing COVID-19 pandemic in Korean children using a modified IPA. The main findings of this study were the following: (1) two factors of diet and two factors of physical activity had high relative importance and low execution among all students, and (2) among both genders, factors such as happiness, sleep environment, sleep hygiene, regular sleep, wearing a mask, and sanitary practice were commonly located on the matrix.

Mask-wearing and sanitary practices led to the best performance because the quarantine policy has become a daily routine due to the ongoing COVID-19 pandemic. In contrast, the lowest performance was for regular sleep activities. Sleep activity was expected to be negatively affected by online classes and excessive exposure to smart media over the past two years, despite the resumption of school attendance.

IPA revealed that the sense of belonging, happiness, trust, and movement activity were located in quadrant I, which can be interpreted as the perceptions being positively executed [17]. However, a previous study conducted with adolescents reported that sanitary practices and diseases were placed in quadrant I [6]. This discrepancy may be due to differences in the study subjects (adolescents versus pre-adolescents). Additionally, the study was conducted during the early COVID-19 pandemic period. Since school has resumed and the COVID-19 policy has been adjusted, interpersonal relationships and physical activity have been restored.

In quadrant II, an adequate quantity of food, dietary habits, sports activities, and regular physical activity were placed. Quadrant II indicates an urgent improvement in the performance level because of its high importance yet low performance [17]. A previous study by Lee et al. [6] reported on mental health in quadrant II. This difference was related to the resumption of school lunches and active physical activity due to full attendance. Although children are aware of the importance of dietary habits and physical activity, it may not be easy to overcome the lifestyle that they have been accustomed to that includes eating instant meals and a sedentary life for over two years. Physical activity and dietary habits are significant factors in maintaining and promoting health [18]; however, there have been reports that the younger generation may not focus on health care [19]. Thus, educational guidance is needed to gradually develop practical dietary habits and physical activity in children.

Quadrant III included preventive care, sleep environment, sleep hygiene, and regular sleep. This result aligns with that of a previous study [6], indicating that excessive and continuous exposure to media such as smartphones, tablets, and laptops caused by online classes seems to have adversely affected sleep [6,16].

Quadrant IV included wearing a mask, sanitary practice, social distancing, prescription control, and disease prevention. The fourth quadrant is perceived as less important than the other quadrants but is practiced more than necessary. Interestingly, a previous study of adolescents revealed that no variables fell into quadrant IV [6], possibly because of the bias toward the first and third quadrants. That is a weakness of the initial IPA analysis. Additionally, wearing a mask, social distancing, hand washing, rapid screening of confirmed cases, and prevention of the spread of the COVID-19 virus have made children aware of the importance of hygiene and treatment. It can be assumed that these five variables were habitualized and practiced despite the adjusted guidelines.

Physical activity performance was higher in boys than in girls. Regarding the performance of physical activity of girls, gender-sensitive physical education classes and programs seem to be needed for girls during the ongoing COVID-19 pandemic [20].

In quadrant I, in which maintaining the current status is necessary, it can be said that happiness is increased after school attendance among both boys and girls. In quadrant II, which requires immediate improvement, both boys and girls did not practice regular physical activity despite its perceived importance. Specifically, in the case of girls, all three variables related to physical activity had lower performance than the perception of importance. Due to the COVID-19 pandemic, physical activity-oriented physical education classes have been replaced by online physical education classes, and school sports club activities within the curriculum and after-school sports clubs have been restricted [21]. Lack of leisure activities lowers awareness and performance of physical activity. Physical activity is important for proper physical growth and immunity enhancement in children, especially girls [6,22].

Three factors related to sleep were found in quadrant III for both boys and girls, which did not require more effort beyond the current level because of low priority. Three factors related to sanitary health were found in Quadrant IV for both boys and girls. It can be assumed that these three variables are habitualized and practiced.

This study has some limitations. First, there were numerous participants. Second, owing to the nature of elementary school students, we were unable to ask many questions. Third, as this study focuses on education and health, some parts are different from the analysis and interpretation of IPA in the marketing field.

## 5. Conclusions

The present study indicates that elementary school students show the highest performance in mask-wearing and sanitary practices. Furthermore, a difference was observed in the performance of physical activity between boys and girls. Therefore, regular and low physical activity and sleep-related factors must be improved. Therefore, the national government, public education institutions, and families should encourage physical activity and appropriate sleep management among children. Future research should investigate the interaction between physical activity and sleep. Furthermore, larger and more diverse populations should be included to ensure the validity and reliability of the improved IPA analysis. Finally, a design that allows the incorporation of a qualitative research method will help in creating broader and deeper interpretations and implications from a multidimensional perspective.

## Figures and Tables

**Figure 1 healthcare-11-00083-f001:**
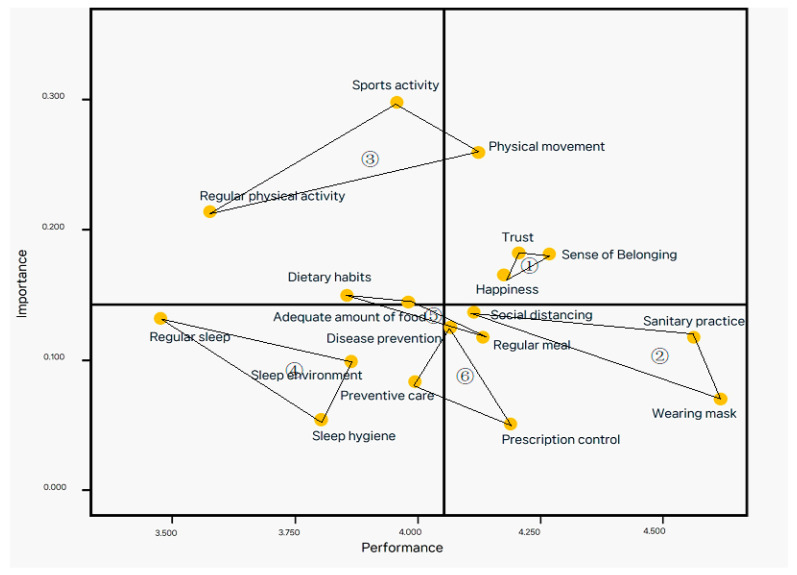
IPA Matrix—Factors.

**Figure 2 healthcare-11-00083-f002:**
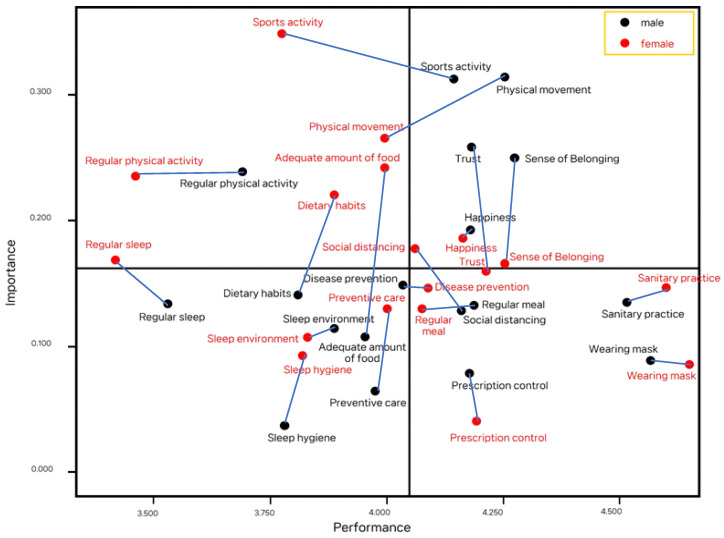
IPA Matrix—Factors based on gender.

**Table 1 healthcare-11-00083-t001:** Exploratory factor analysis and reliability analysis.

Variable	Hygience	Mental Health	Sleep	Diet	Disease	Physical Activity
Happiness	0.057	0.782	0.092	0.099	0.060	0.097
Sense of belonging	0.076	0.825	0.060	0.116	0.101	0.092
Trust	0.074	0.809	0.108	0.120	0.040	0.106
Disease prevention	0.091	0.133	0.007	0.119	0.765	0.099
Prescription control	0.122	0.029	0.059	0.095	0.811	−0.079
Preventive care	0.132	0.037	0.103	0.053	0.779	0.075
Sports activity	0.003	0.092	0.020	0.058	0.103	0.811
Physical movement	0.069	0.115	0.083	0.137	0.037	0.796
Regular physical activity	0.115	0.080	0.144	0.071	−0.044	0.740
Regular sleep	0.032	0.073	0.796	0.132	0.112	0.176
Sleep hygiene	0.122	0.094	0.823	0.103	0.069	0.035
Sleep environment	0.052	0.092	0.741	0.181	0.000	0.053
Regular meal	0.076	0.175	0.247	0.694	0.188	0.044
Dietary habits	0.99	0.092	0.127	0.795	0.081	0.173
Adequate quantity of food	0.152	0.120	0.119	0.821	0.056	0.092
Wearing mask	0.848	0.047	0.023	0.165	0.132	0.015
Social distancing	0.698	0.047	0.023	0.165	0.132	0.015
Sanitary practice	0.852	0.058	0.022	0.145	0.125	0.054
Eigen value	2.094	2.088	2.048	1.993	1.987	1.985
Explained (%)	11.634	11.602	11.378	11.074	11.037	11.030
Cumulative (%)	11.634	23.236	34.614	45.689	56.725	67.756
Cronbach’s α	0.744	0.771	0.751	0.765	0.725	0.720

**Table 2 healthcare-11-00083-t002:** Participants’ demographic characteristics.

		Number	Percentage (%)
Gender	Boys	481	47.8
	Girls	525	52.2
Total		1006	100

**Table 3 healthcare-11-00083-t003:** Analysis of importance and performance by categories.

Variables		Importance	Performance
① Mental Health	Happiness	0.188	4.170
	Sense of belonging	0.207	4.262
	Trust	0.208	4.198
② Disease management	Disease prevention	0.142	4.063
	Prescription control	0.058	4.183
	Preventive care	0.095	3.988
③ Physical activity	Sports activity	0.340	3.951
	Physical movement	0.296	4.118
	Regular physical activity	0.244	3.571
④ Sleep	Regular sleep	0.151	3.470
	Sleep hygiene	0.061	3.798
	Sleep environment	0.112	3.858
⑤ Diet	Regular meal	0.134	4.128
	Dietary habits	0.170	3.850
	Adequate quantity of food	0.165	3.975
⑥ Sanitary Health	Wearing mask	0.080	4.612
	Social distancing	0.155	4.108
	Sanitary practice	0.133	4.558
Mean		0.163	4.048

**Table 4 healthcare-11-00083-t004:** Distribution of health perception by variables.

Quadrant	Criteria	Variables
Quadrant I	Importance ↑ Performance ↑	Sense of belonging, Happiness, Trust, Physical movement
Quadrant II	Importance ↑ Performance ↓	Adequate quantity of food, Dietary habits, Sports activity, Regular physical activity
Quadrant III	Importance ↓ Performance ↓	Preventive care, Sleep environment, Sleep hygiene, Regular sleep
Quadrant IV	Importance ↓ Performance ↑	Wearing a mask, Sanitary practice, Social distancing, Prescription control, Disease prevention

**Table 5 healthcare-11-00083-t005:** The difference in performance between boys and girls.

Categories	Gender	N	M	SD	SE	*t*	*p*
Mental health	Boys	481	4.211	0.679	0.031	0.027	0.979
	Girls	525	4.210	0.684	0.030		
Disease management	Boys	481	4.062	0.679	0.031	−0.719	0.472
	Girls	525	4.092	0.631	0.028		
Physical activity	Boys	481	4.028	0.826	0.038	5.221	0.000 ***
	Girls	525	3.744	0.901	0.039		
Sleep	Boys	481	3.733	0.959	0.044	0.780	0.435
	Girls	525	3.687	0.891	0.039		
Diet	Boys	481	3.983	0.876	0.040	−0.062	0.950
	Girls	525	3.986	0.828	0.036		
Sanitary health	Boys	481	4.414	0.723	0.033	−0.521	0.603
	Girls	525	4.437	0.634	0.028		

*** Significance difference between genders (*p* ≤ 0.01).

**Table 6 healthcare-11-00083-t006:** Analysis of the importance and performance of children’s health perception between boys and girls.

Categories	Variables	Boys		Girls	
		Importance	Performance	Importance	Performance
Mental health	Happiness	0.192	4.179	0.186	4.162
	Sense of belonging	0.250	4.272	0.165	4.253
	Trust	0.258	4.181	0.160	4.213
Disease management	Disease prevention	0.149	4.035	0.147	4.088
	Prescription control	0.079	4.177	0.040	4.189
	Preventive care	0.06	3.975	0.130	4.000
Physical activity	Sports activity	0.313	4.143	0.349	3.775
	Physical movement	0.315	4.252	0.266	3.996
	Regular physical activity	0.239	3.690	0.236	3.461
Sleep	Regular sleep	0.134	3.530	0.169	3.415
	Sleep hygiene	0.037	3.780	0.093	3.815
	Sleep environment	0.115	3.888	0.107	3.830
Diet	Regular meal	0.133	4.187	0.130	4.074
	Dietary habits	0.141	3.809	0.221	3.888
	Adequate quantity of food	0.107	3.952	0.242	3.996
Sanitary health	Wearing mask	0.089	4.570	0.085	4.651
	Social distancing	0.128	4.160	0.178	4.061
	Sanitary practice	0.135	4.514	0.148	4.598
Average		0.160	4.072	0.170	4.026

**Table 7 healthcare-11-00083-t007:** Distribution of health perception by gender.

Quadrant	Criteria	Gender	Variables Distribution
Quadrant I	Importance ↑ Performance ↑	Boys	Happiness, Trust, Physical movement, Sports activity
		Girls	Happiness, Social distancing
Quadrant II	Importance ↑ Performance ↓	Boys	Regular physical activity
		Girls	Regular physical activity, Physical movement, Sports activity, Adequate quantity of food, Dietary habits
Quadrant III	Importance ↓ Performance ↓	Boys	Sleep environment, Sleep hygiene, Regular sleep, Disease prevention, Preventive care, Dietary habits, Adequate quantity of food,
		Girls	Sleep environment, Sleep hygiene, Regular sleep
Quadrant IV	Importance ↓ Performance ↑	Boys	Wearing a mask, Sanitary practice, Prescription control, Social distancing, Regular meal
		Girls	Wearing a mask, Sanitary practice, Happiness, Trust, Prescription control, Sense of belonging

## Data Availability

The data presented in this study are available upon request from the corresponding author. The data were not publicly available because of the protection of personal information.

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
