# Peer review of "An Analysis of Health Perceptions and Performance in Elementary Students in Korea during the ongoing COVID-19 Pandemic"

_healthcare, 2022, doi:10.3390/healthcare11010083_

Round 1
Reviewer 1 Report
Allow me to congratulate you for the work you have done. However, let me list a few comments for you:
Page 1 – Abstract: It is not clear from the abstract which variables have been valued.
Page 2 – Introduction: I encourage you to redo the second part of the introduction because it is not entirely clear.
Page 2 – Participants: You should describe in this section which sample this study is aimed at and the inclusion and exclusion criteria. You should not present data on how many people you have recruited since that belongs to the results section.
Page 2-3 – Methods: Please indicate other data as period of time you used to collect the sample, if informed consent was given to them, if permission was requested from the ethics committee (especially because they are minors), or if the study is registered previously or posteriori.
Page 3 – Point 2.3: I think this point better in results.
Page 4 – Results: Firstly, I recommend a flow diagram of the intervention and incorporate demographic data of the participants.
Page 6 – Table 5: Please indicate under the table the meaning of the acronyms used.
Page 7 – Figure 2: Please try to improve the quality of the image.
Page 8 – Discussion 1st paragraph: This paragraph should answer the objectives stated in the introduction.
Page 9 – Discussion: It would be interesting if they provided the detected limitations that the work itself entails.
Page 10 – Conclusion: The conclusion must be improved. Don’t forget to answer in this point the main objectives of your study.
Author Response
Thank you for your opinion. We marked the revised contents in yellow fluorescent color. Thank you for the good judgment
Page 1 – Abstract: It is not clear from the abstract which variables have been valued.
-> Along with the purpose of the study, specific factors related to health awareness were modified and supplemented.
Page 2 – Introduction: I encourage you to redo the second part of the introduction because it is not entirely clear.
-> The second paragraph of the introduction has been corrected.
Page 2 – Participants: You should describe in this section which sample this study is aimed at and the inclusion and exclusion criteria. You should not present data on how many people you have recruited since that belongs to the results section.
->As suggested by the reviewer, except for the survey number of people, the contents were presented in the sample to be studied and modified
Page 2-3 – Methods: Please indicate other data as period of time you used to collect the sample, if informed consent was given to them, if permission was requested from the ethics committee (especially because they are minors), or if the study is registered previously or posteriori.
->Since we received permission from the Ethics Committee, we have revised and supplemented the contents related to irb.
Page 3 – Point 2.3: I think this point better in results.
-> Thank you for your comments.
Page 4 – Results: Firstly, I recommend a flow diagram of the intervention and incorporate demographic data of the participants.
->The contents were added and supplemented as suggested by the reviewer
Page 6 – Table 5: Please indicate under the table the meaning of the acronyms used.
->It means that there was a significant difference between men and women in physical activity by less than 0.01.
Page 7 – Figure 2: Please try to improve the quality of the image.
->Image has been corrected.
Page 8 – Discussion 1st paragraph: This paragraph should answer the objectives stated in the introduction.
->Corrected the content of the first paragraph of the discussion.
Page 9 – Discussion: It would be interesting if they provided the detected limitations that the work itself entails.
-> Limitations are presented at the end of the discussion on page 9.
Page 10 – Conclusion: The conclusion must be improved. Don’t forget to answer in this point the main objectives of your study.
->The conclusions were revised and presented again.

Reviewer 2 Report
The paper has several issues.
1. When was the study conducted and why? How does it fit the journal scope and mission?
2. Why was this sample recruited and not others? What about the influence of adult social networks and the environment>
3. How was reliability and validity estimated for measures?
4. What is the relevance of the study now that we are emerging out of the acute phase of the pandemic?
5. What are the implications for practice, policy, and prevention?
6. What will be the future directions for research?
Author Response
Thank you for your opinion. We marked the revised contents in yellow fluorescent color. Thank you for the good judgment . Attached is a certificate that our research team has been edited by a professional English correction team. Thank you.
- When was the study conducted and why? How does it fit the journal scope and mission?
-> Reflecting the review opinion, the research period was added to the research participant area.
2. Why was this sample recruited and not others? What about the influence of adult social networks and the environment
-> The importance of cultivating a healthy body and mind of elementary school students, and the need for life skills to lead a healthy life in daily life through practice was suggested.
3. How was reliability and validity estimated for measures?
-> Added assessment of reliability and validity of measurements as suggested by the reviewer
4. What is the relevance of the study now that we are emerging out of the acute phase of the pandemic?
-> It is meaningful in the results that the indicator for ‘physical activity’, which was a problem in the acute phase of the COVID-19 pandemic, gradually showed a recovery in some factors as the acute phase eased. It is meaningful in the results that the indicator for ‘physical activity’, which was a problem in the acute phase of the COVID-19 pandemic, gradually showed a recovery in some factors as the acute phase eased.
5. What are the implications for practice, policy, and prevention?
-> As revealed in the results of this study, continuous attention to sleep is necessary. In particular, in schools, continuous treatment for physical activities in everyday life is required in addition to class-oriented physical activities.
6. What will be the future directions for research?
-> In the case of Korea, since full school attendance was implemented, it is necessary to follow up and study the health aspects of children in the situation of full school attendance. In addition, there is a need for research in relation to physical activity and health in relation to children's injuries in schools, which have recently occurred frequently.
